# Study on Urban Expansion Using the Spatial and Temporal Dynamic Changes in the Impervious Surface in Nanjing

**Yanping Qian [2]**  **and Zhen Wu [1,\*]**

1  School of Architecture, Nanjing Tech University, Nanjing 210000, China
2  College of Landscape Architecture, Nanjing Forestry University, Nanjing 210000, China; qianyanping705@163.com
\*  Correspondence: 201910006529@njtech.edu.cn

**Abstract:** Impervious surface area is a key factor affecting urbanization and urban environmental quality. It is of great significance to analysis timely and accurately the dynamic changes of impervious surface for urban development planning. In this study, we use a comprehensive method to extract the time series data on the impervious surface area (ISA) from the multi-temporal Landsat remote sensing images with a high overall accuracy of 90%. The processes and mechanisms of urban expansion at different political administration and direction level in the Nanjing metropolitan area are investigated by using the comprehensive classification method consisting of minimum noise fraction, linear spectral mixture analysis, spectral index, and decision tree classifiers. The expansion of Nanjing is examined by using various ISA indexes and concentric regression analyses. Results indicate that the overall classification accuracy of ISA is higher than 90%. The ISA in Nanjing has dramatically increased in the past three decades from 427.36 km$^2$ to 1780.21 km$^2$ and with a high expansion rate of 0.48 from 2000 to 2005. The city sprawls from monocentric to urban core with multiple subcenters in a concentric structure, and the geometric gravity center of construction land moves southward annually. The stages of urbanization in different district levels and the dynamic changes in different direction levels are influenced by the topographic and economic factors.

**Keywords:** urban expansion; impervious surface area; Nanjing

## 1. Introduction

Development of a city during rapid urbanization is often at the expense of natural landscape pervious surfaces which are replaced with an impervious surface composed of mixed clay, asphalt, stone, and metal [1]. The transformation of the earth's surface from non-urban land to urban land becomes an irreversible process [2,3], thereby generating serious problems, such as farmland loss, heat island effect, flood hazard, and habitat fragmentation [4–13].

In recent years, many studies have focused on examining the urban land cover and its dynamic changes using the remote sensing (RS) method [14–17]. However, urban land cover extraction from RS images results in poor accuracy because an urban land is usually a mosaic of different land covers, such as plants, soils, buildings, and water [18,19]. Impervious surface area (ISA) is an essential component of urban landscapes and is the artificial surface that cannot be penetrated by water because of the composition of man-made construction materials in structures, such as road and building roofs [18]. ISA can truly reflect the expansion of urban construction land and can be extracted from RS images [13].

Ridd proposed the vegetation–impervious surface–soil conceptual model to explain urban composition [20]. He regarded urban surface as a combination of vegetation, impervious surface, and

soil [20]. Wu and Murray applied the conceptual model of fixed end-member to the Landsat Thematic Mapper Plus (ETM+) image, and they developed the real and operable linear spectral mixture analysis (LSMA) [21]. Various methods, such as spectral mixture analysis, multiple end-member spectral mixture analysis, and ISA index, have been used to improve the precision of ISA extraction [21–26]. RS data, such as QuickBird, DMSP-OLS, Terra MODIS, and so on, have also been used to improve accuracy [27–31].

Developing countries, especially China, have become the main driving force of global urbanization in recent years [32–34]. The number of large-sized and medium-sized cities are increasing rapidly, and the scale of urban construction is growing. China has experienced an acceleration of urban expansion from the late 1980s [33]. In addition, 70% population of Chinese will live in urban areas by 2030, and 200 million rural immigrants will be received by Chinese cities [35]. As a result, a sharp increase in ISA will occur.

This study uses a large city, Nanjing, as an example. The sprawl of the city and each district is measured using impervious surface, and the consequences of city expansion are analyzed. Nanjing is the old capital of six dynasties in Chinese history and the capital of Jiangsu Province [36]. The urban expansion and development trajectories of Nanjing differ from those of globalizing first-tier metropolitan cities in China in recent years, such as Beijing and Shanghai [37]. Nanjing is a second-tier city with a rapidly growing global economy and is a representative site of research [38].

Some studies have been conducted on the spatiotemporal patterns of urban expansion by monitoring the dynamic changes in ISA [37,39–43]. The current research not only analyzes the status of ISA development in the entire city but also quantifies and compares spatiotemporal patterns of ISA and urban expansion details in different directions and districts. The impacts of the economy on different district development phases are also analyzed. Therefore, this study mainly aims to

(1) Use a comprehensive method to extract ISA from Landsat images with an overall accuracy of 90%;
(2) Map locations, density, the geometric center of gravity, and extents of ISA in Nanjing;
(3) Quantify spatiotemporal patterns of urban expansion in the entire city, and
(4) Comprehensively compare similarities and differences in the development phases of urbanization under different direction and district levels.

This type of research provides a method to improve the accuracy of urban expansion research data extracted from RS images and provide new understanding on the interactions of socioeconomic systems, different development phases, expansion mechanisms, and urbanization patterns from the perspective of direction, district levels, and the entire city. The other value of this study is to better understand the details of urban expansion in Nanjing, and to provide valuable insights for improving urban planning, management, and sustainability.

## 2. Materials and Methods

### 2.1. Study Area

Nanjing is a city with a 2000 years history and was an ancient Chinese capital [36]. Nanjing is the current capital of Jiangsu Province [38]. Thus, Nanjing is the center of politics, economy, and culture of the province and is the core area of the Yangtze River economic belt [44]. Its level of economic development is at the forefront in China [44].

Nanjing metropolis includes 11 districts, namely, Xuanwu, Qinhuai, Gulou, Jianye, Yuhuatai, Qixia, Pukou, Luhe, Jiangning, Lishui, and Gaochun, with a total area of 6587.02 km$^2$ [45]. Its economic growth accelerated after the economic reform, especially in the 1990s. During this period, it was transformed from a medium-sized city to a megacity. Its Gross Domestic Product (GDP) and the municipal population has grown from RMB 16 billion and 5 million people in 1990 to RMB 9720 billion and 8 million people in 2015 [45] (Figure 1).

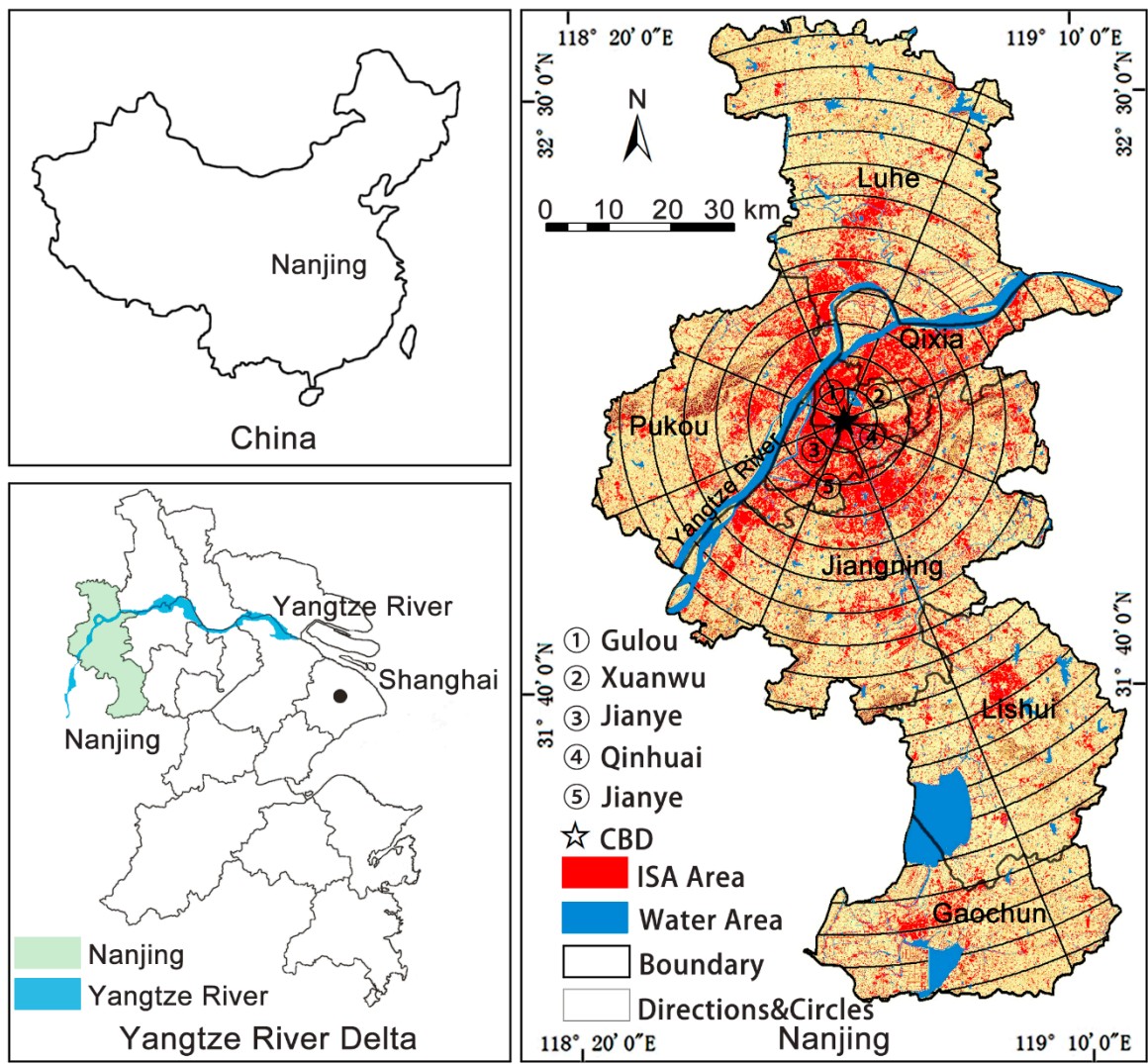

**Figure 1.** Location of the study area.

## 2.2. Data Collection and Preprocessing

RS data used in this research was mainly from the Landsat Thematic Mapper (TM), Enhanced ETM+, and Operational Land Imager from 1990, 1995, 2000, 2005, 2010, and 2017 (Path: 120, Row: 38) (Table 1). The digital elevation model (DEM) data came from the ASTER Global DEM (GDEM) in Nanjing. The Environmental Systems Research Institute (ESRI) Shapefile of Nanjing administrative units was used. The images were georeferenced in the Universal Transverse Mercator system (WGS84 datum, Zone51 North), and Google Earth images from 1990, 1995, 2000, 2005, 2010, and 2017 were used as the reference data for evaluating ISA accuracy.

Population and GDP data of Nanjing megacity were collected from Nanjing Statistical Yearbook, which is published by China Statistics Press. Ancillary data such as urban planning and management was also collected for examining the different influences of urban development policies on urbanization patterns and rates.

**Table 1.** Summary of data used in research.

| Data Types | Datasets | | |
|---|---|---|---|
| | Time | Path/row | Image type |
| Remote sensing data | 1990-7-11 | Landsat 5-TM | 120/38 |
| | 1995-10-13 | Landsat 5-TM | 120/38 |
| | 2000-6-12 | Landsat 5-TM | 120/38 |
| | 2005-4-7 | Landsat 5-TM | 120/38 |
| | 2010-4-5 | Landsat 7-ETM | 120/38 |
| | 2017-7-21 | Landsat 8-OLI | 120/38 |
| DEM data | ASTER GDEM with 30m spatial resolution | | |
| Boundary files | Nanjing administration boundary shape data | | |
| GDP data | GDP data in 1995, 2000, 2005, 2010, and 2015 year at district level in Nanjing | | |
| Ancillary data | Urban planning and policies of Nanjing | | |

*2.3. Mapping ISA Distribution*

The proposed comprehensive classification framework for improved LSMA consisted of minimum noise fraction (MNF), LSMA, spectral index, and decision tree classifiers.

Figure 2 illustrates the major steps of the framework. The major steps are explained in detail below:

(1) Basic RS image operations, such as atmospheric correction and radiometric calibration, were used. ISO unsupervised classification and spectral index of MNDWI were calculated through ENVI 5.2 [46,47].

(2) MNF transform is a tool for determining the number of bands in the image data, separating the noise in the data, and reducing the computing demand [21]. This method can effectively eliminate noise and reduce the dimension of an image. We used the MNF method to extract good-quality end-members. High albedo, low albedo, and vegetation end-members were extracted on the basis of the geometric vertices in a two-dimensional scatter graph [48].

(3) Three end-members were used as the ROIs to produce fractional images by using the LSMA approach [48].

(4) Three fractional images, namely, high-albedo, low-albedo, and green vegetation fraction images, with the Iterative Selforganizing Data Analysis Techniques Algorithm (ISODATA) were used in the decision tree classifier to produce ISA and other data.

(5) Post processing was conducted, and the results of ISA accuracy were evaluated [49].

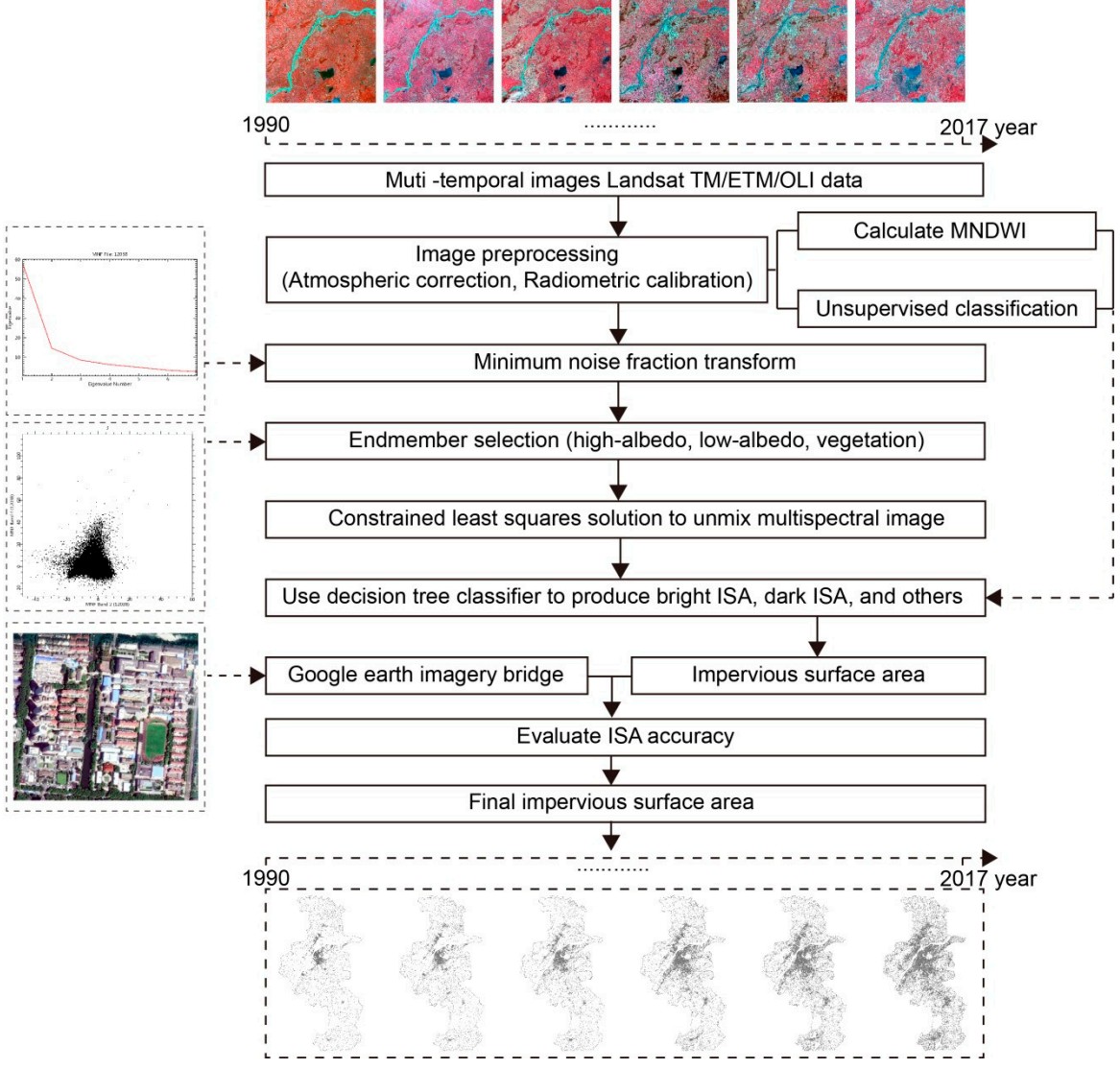

**Figure 2.** Flowchart for mapping impervious surface distribution.

## 2.4. Dynamic Changes in ISA

### 2.4.1. Density of Dynamic Change in ISA

We mapped the density of ISA to reflect the characteristics of its expansion in time and space. This can intuitively explain the time series of ISA distribution characteristic in the entire city and can illustrate the distribution of human activities. We transformed the original 30 m-sized ISA raster images into the 1 km-sized ISA raster images by using the aggregate method on ArcGIS software [50]. As a result, the scale and direction of the ISA transformation from the rural areas to the urban areas could be determined.

The following formula can be used to calculate the density of ISA:

$$\text{Density of ISA } DOI = \frac{ISA}{TA}. \tag{1}$$

where *DOI* is the density of ISA in the study area, *ISA* is the area of impervious surface in the study area, and *TA* is the total area in the study area.

### 2.4.2. Center of Gravity of Dynamic Change in ISA

The geometric center of gravity of ISA distribution was measured by ArcGIS to reflect the overall situation of urban expansion in Nanjing. The center of gravity model is composed of 11 district units, and the center of gravity model of ISA is as follows:

$$\text{Gravity center of ISA } ISA_i(x,y) = \frac{\sum_i^n (I_i Q(x_i, y_i))}{\sum_i^n I_i}, \qquad (2)$$

where $ISA_i(x,y)$ are the geographic coordinates of the center of gravity of ISA in Nanjing, $I_i$ is the area of the impervious surface in different districts, and $Q(x_i, y_i)$ are the geographic coordinates of the center of gravity of ISA in different districts in Nanjing.

### 2.4.3. Expansion Rate and Area of Dynamic Change in ISA

Some indexes of ISA change were used to analyze the entire city and its districts. By analyzing the ISA change data in time and space, the expansion of a city can be described. The following variables are calculated to analyze the dynamic change in ISA [51]. The first two formulas are mainly analyzed from the area of ISA, and the latter formulas are mainly analyzed from the growth efficiency of ISA.

$$\text{Overall ISA changed area} \quad OICA = ISA(t2) - ISA(t1) \qquad (3)$$

$$\text{Annual expansion area} \quad AEA = [ISA(t2) - ISA(t1)]/(t2 - t1) \qquad (4)$$

$$\text{Expansion rate} \quad ER = [ISA(t2) - ISA(t1)]/ISA(t1) \qquad (5)$$

where $ISA(t)$ is the area of ISA in different time, $t1$ is the prior time of $t2$.

### 2.4.4. Dynamic Change in ISA in Different Directions

One urban expansion model assumes that urban expansion is a concentric circle structure [44]. This model indicates that a city takes the Central Business District (CBD) as its expansion circle center and expands from the center to suburbs in the form of concentric circles [52,53].

On the basis of this theory, we created the concentric buffer zone structure of the ISA in ArcGIS to analyze the city expansion in different directions.

We defined Xinjiekou CBD as the concentric circle center of Nanjing on the basis of previous research experience [52]. This CBD is located at the intersection of Hanzhong Road and Zhongshan Road. Meanwhile, 5 km buffer zone intervals were created to cover the entire city [44]. An equal interval of 22.5 degrees for each direction was used, and the study area and the concentric circles were divided into eight parts: east, southeast, south, southwest, west, northwest, north, and northeast (Figure 1). The density and expansion rate of ISA were calculated using this method to investigate the spatial expansion of the city in different directions.

### 2.4.5. Impacts of Topography on Dynamic Change in ISA

Nanjing is in a hilly region [36]. Thus, the impact of terrain on its urban expansion should be investigated. In explaining the relationship between the impact of topography and city expansion, different interval ranges of elevation and slope were used as statistical indicators to describe the distribution and dynamic change in ISA in different topographies. In this case, the elevation was divided into six levels, and the slope was divided into four levels.

### 2.4.6. Relationship between ISA Dynamics and GDP in Different Districts

Economic development has directly promoted the process of urbanization and determined the level of urbanization [3,54]. However, the relationship between economic and ISA changes at the district level has not been investigated. In this study, we used the regression models to explore the

relationship between the GDP data and the ISA change data from 11 districts in Nanjing. We aimed to find the different development phases of five urban core districts and six suburban districts in Nanjing and to develop effective strategies of urban planning on different urban expansion patterns in district level.

## 3. Results

### 3.1. Dynamic Change in ISA in the Entire City

The accuracy assessment of the confusion matrix showed that the overall accuracy of the ISA data in Nanjing was more than 90% in 2010 and 2017 (Table 2). The producer's and user's accuracies in 2010 were 89.0% and 94.5%, respectively. The producer's and user's accuracies in 2017 were 92.0% and 95.5%, respectively. According to previous researchers, the results in other time series were in similar accuracy.

**Table 2.** Accuracy assessment of impervious surface mapping result.

| | 2010 | | | | 2017 | | | |
|---|---|---|---|---|---|---|---|---|
| | ISA | Others | Producer's accuracy (%) | User's accuracy (%) | ISA | Others | Producer's accuracy (%) | User's accuracy (%) |
| ISA | 89 | 11 | 89.0 | 89.0 | 92 | 8 | 91.1 | 92.0 |
| Others | 11 | 189 | 94.5 | 94.5 | 9 | 191 | 96.0 | 95.5 |
| Overall accuracy (%) | | | 92.7 | | | | 92.6 | |

As shown in Figure 3, the density of ISA distribution in Nanjing has drastically changed in the past 30 years. The density of ISA increased from the urban core to the suburban, and the ISA expansion was separated into two parts by the Yangtze River. After 2000, the ISA located in the northern part of the Yangtze River accelerated and expanded. The urban expansion was marked by two axes in the east–west and in the north–south.

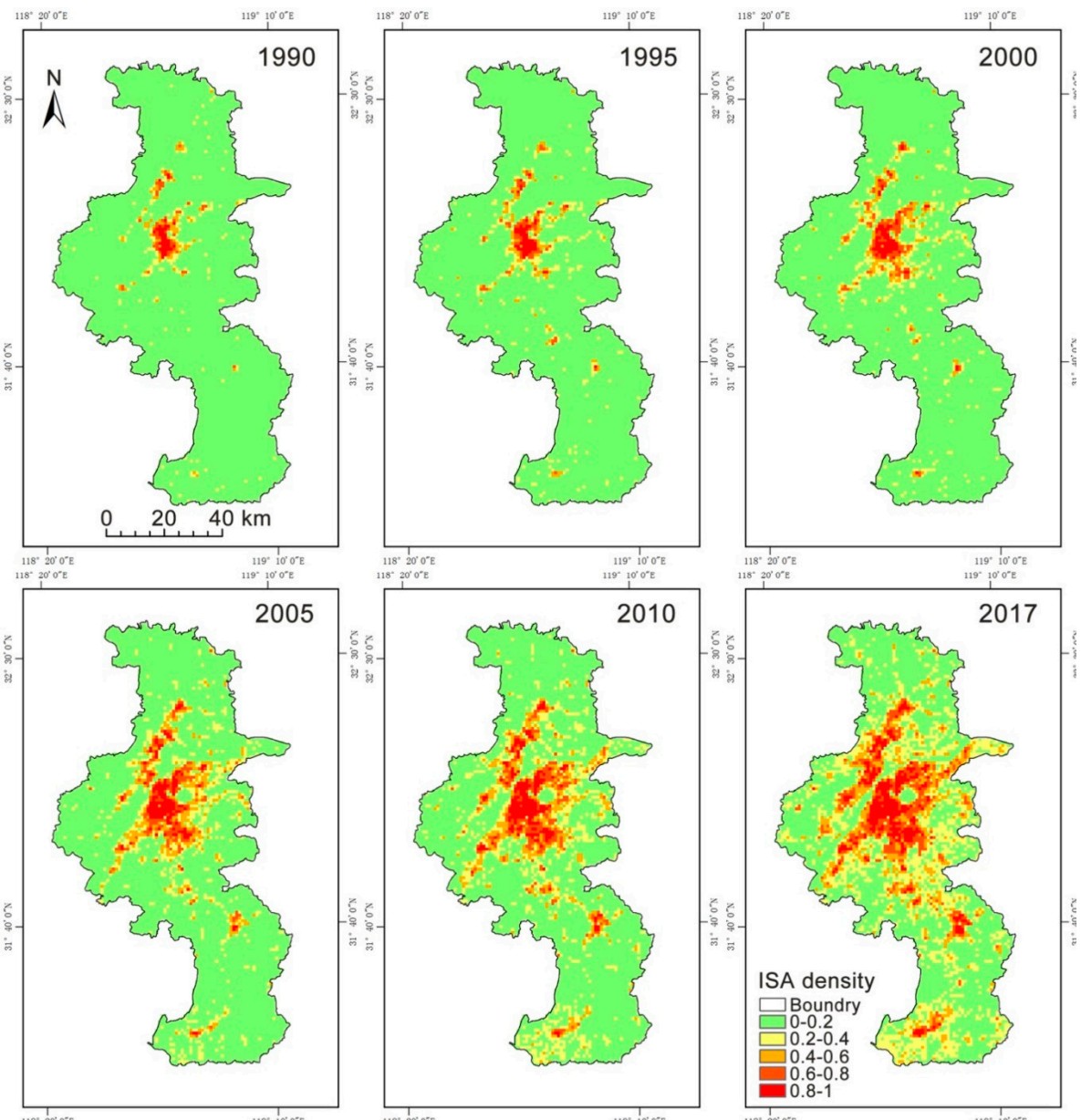

**Figure 3.** Density of dynamic change in impervious surface area (ISA) in Nanjing from 1990 to 2017.

The transitions in the geometric gravity center of ISA change can explain the overall situation of urban expansion. From 1990 to 2017, the geometric center of gravity of ISA generally moved southward annually. This transition was divided into two stages. In the first stage, from 1990 to 1995, the center of gravity of ISA gradually moved to southwest; in the second stage, from 1995 to 2017, the center of gravity of ISA gradually shifted to the southeast. The southern expansion of Nanjing was obvious from 2005 to 2010. However, in the past years from 2010 to 2017, the distance between the geometric center of gravity of ISA of the two years became small. Therefore, the construction of ISA increased in the northwest, and the expansion became balanced between the north and south directions (Figure 4).

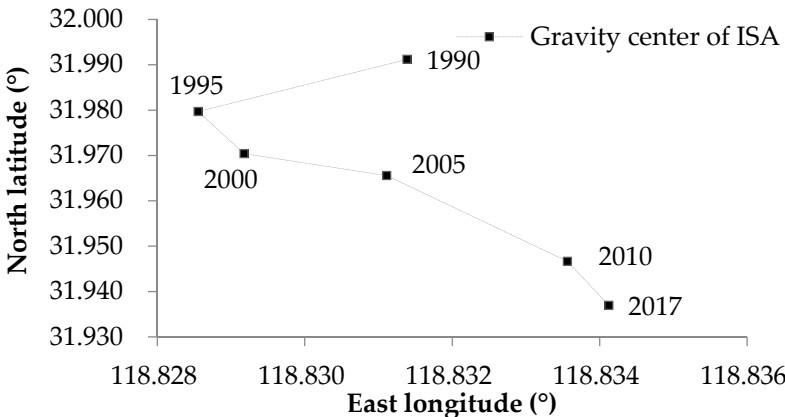

**Figure 4.** Center of gravity of dynamic change in ISA in Nanjing from 1990 to 2017.

Overall, the ISA in Nanjing metropolis increased rapidly from 427.36 km$^2$ in 1990 to 1780.21 km$^2$ in 2017 (Table 3).

**Table 3.** Impervious surface area (ISA) data in Nanjing from 1990 to 2017.

| Year | 1990 | 1995 | 2000 | 2005 | 2010 | 2017 |
|------|------|------|------|------|------|------|
| ISA (km$^2$) | 427.36 | 545.57 | 668.72 | 987.58 | 1271.93 | 1780.21 |

With regard to area change, the increasing amount of ISA and the annual expansion area reached the maximum in the period of 2010 to 2017; the period of 2000 to 2005 was in second place; the area change in other periods was relatively low. In terms of rate change, the expansion rate reached 0.48 in the period 2000 to 2005, respectively; on the contrary, the rates only reached 0.23 in the period 1995 to 2000. The calculation results of urban expansion showed that, before 2000, the urban expansion of Nanjing was relatively slow; after 2000, Nanjing rapidly developed, especially in the periods 2000 to 2005 and 2010 to 2017 (Table 4).

**Table 4.** Analysis of dynamic change in ISA in Nanjing from 1990 to 2017.

|  | 1990–1995 | 1995–2000 | 2000–2005 | 2005–2010 | 2010–2017 |
|---|-----------|-----------|-----------|-----------|-----------|
| Overall ISA changed area (km$^2$) | 118.22 | 123.15 | 318.86 | 284.34 | 508.29 |
| Annual expansion area (km$^2$/year) | 23.64 | 24.63 | 63.77 | 56.87 | 72.61 |
| Expansion rate | 0.28 | 0.23 | 0.48 | 0.29 | 0.40 |

*3.2. Dynamic Change in ISA in District Level*

Planners usually distinguish the 11 districts of Nanjing as two types: urban core and suburban districts (Figure 1). Urban core districts include Xuanwu, Qinhuai, Gulou, Jianye, and Yuhuatai, while suburban districts include Qixia, Pukou, Luhe, Jiangning, Lishui, and Gaochun.

Figure 5(a1,a2) indicates obvious differences between the urban core and suburban districts in terms of ISA distribution. At the total tendency of ISA amount, the amount of ISA in all urban core districts was below 60 km$^2$, and the trend of ISA growth gradually slowed down because of the limited space in the urban districts. The ISA amount was over 150 km$^2$, and the trend in ISA growth steadily increased in the suburban districts compared with those in the urban core districts. Although the trend in ISA growth in the urban core districts slowed down, the amount of ISA in Jianye and Yuhuatai in the urban core increased relatively more than that in others. Among the suburban districts, the amount of ISA in Jiangning and Luhe was relatively large owing to the urban development and the large area of their own administration.

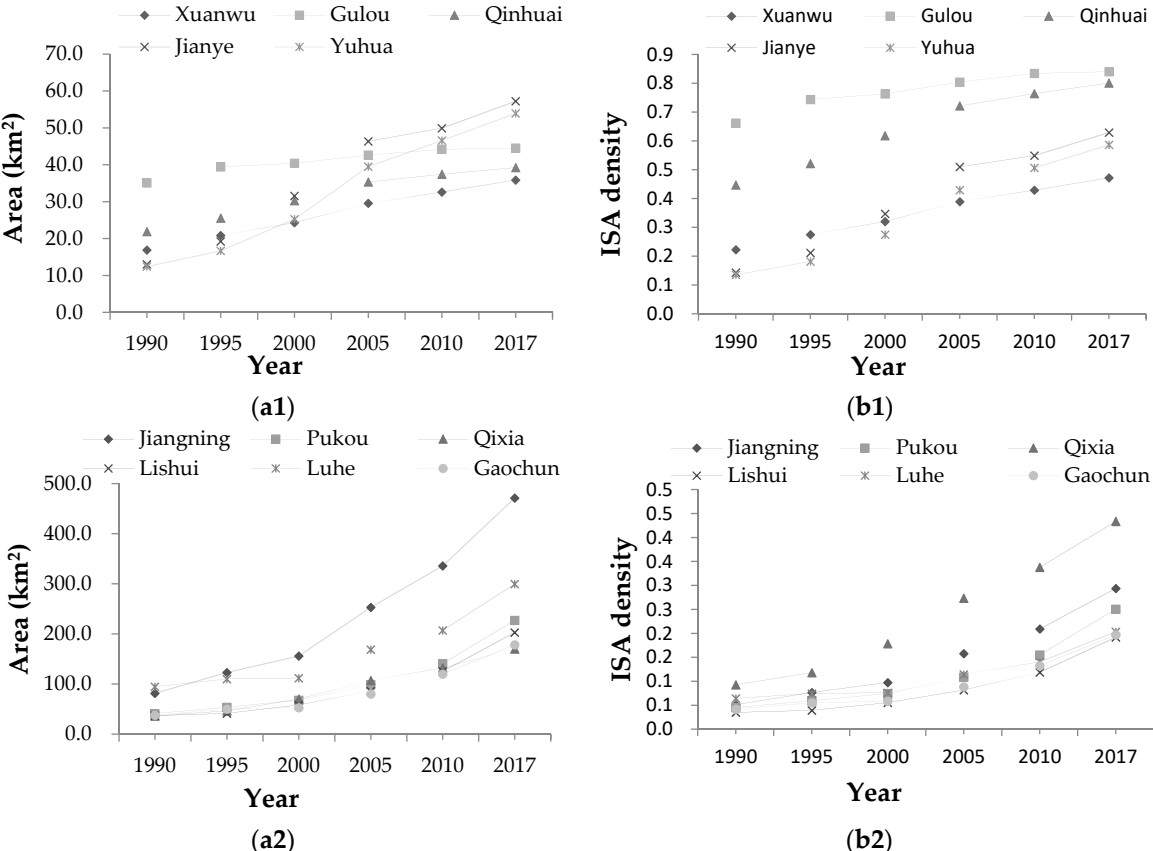

**Figure 5.** Amount and density of ISA in different districts in Nanjing from 1990 to 2017. (Note: (**a1**) Amount of ISA in urban core districts; (**a2**) Amount of ISA in suburban districts; (**b1**) Density of ISA in urban core districts; (**b2**) Density of ISA in suburban districts).

Figure 5(b1,b2) indicate that the urban core districts presented higher ISA densities than the suburban districts. The density of ISA in the urban core districts was more than 40%. Among them, the ISA density of Gulou and Qinhuai were the largest. By contrast, Xuanwu exhibited a low density owing to the Zijin mountains and the Xuanwu Lake, which are ecologically protected areas. In the future development of the urban core districts, Jianye and Yuhuatai will further develop to be the main areas of city construction.

In the suburbs, the density of ISA was less than 40%, except in Qixia. The development trend of Qixia was gradually near that of Jianye and Yuhuatai because its geographical location is near the core of the city. Gaochun, Luhe, and Lishui presented a low density of ISA because of their large agricultural and mountainous areas.

The density of ISA in other suburban districts was around 20%–30%. However, the development trends of the ISA density showed an accelerated increase in the period 2005 to 2017. The density will grow rapidly in the future. The results also coincided with the actual situation. In particular, three suburban cores were formed: Xianling suburban core in Qixia District, Dongshan suburban core in Jiangning District, and Jiangbei suburban core in Pukou District.

Figure 6 indicates that the increase of ISA in suburban districts was more than that in urban core districts. In recent years, the increase in urban area is below 1.5 km$^2$/year, and the increase in the suburb is over 5 km$^2$/year. The annual expansion area in Jianye and Yuhuatai increased more than that in three other districts; the annual expansion area development of the three districts in the city core was consistent. Jiangning exhibited the highest value of the annual expansion area. The construction of Jiangning and Lishui has been growing rapidly since 1990, and the construction of Qixia has slowed down since 2005.

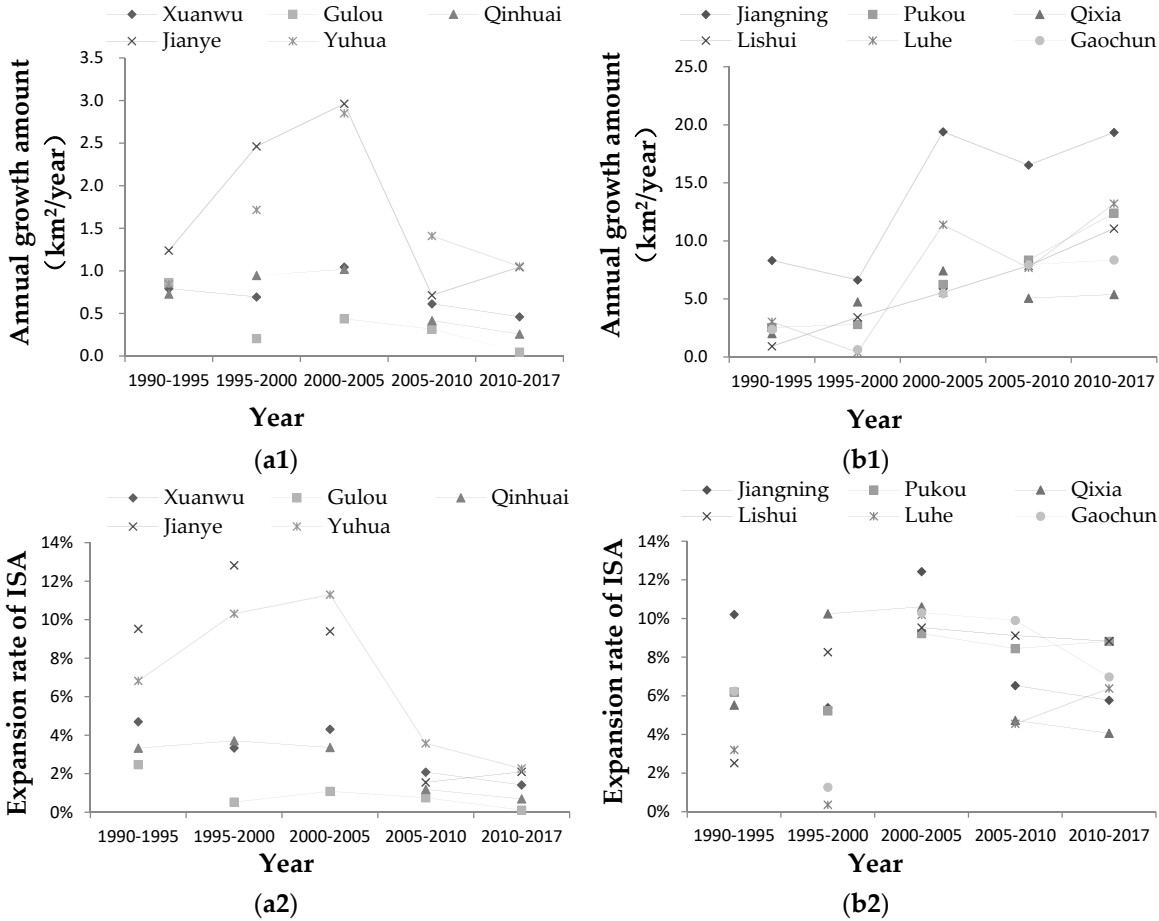

**Figure 6.** Annual growth amount and rates of ISA in different districts in Nanjing from 1990 to 2017. (Note: (**a1**) Annual growth amount of ISA in urban core districts; (**a2**) Annual growth amount of ISA in suburban districts; (**b1**) Annual growth rate of ISA in urban core districts; (**b2**) Annual growth rate of ISA in suburban districts).

After 2005, obvious differences were found in the development of the urban core and suburban districts. The growth point of ISA shifted from the urban area to the suburban, and the suburb entered the main period of development.

Figure 6(b2) shows that, in the period 1990 to 2005, Jianye and Yuhuatai were rapidly developed. However, the annual expansion growth rate of all the urban core districts slowed down after 2005, and the trend of the urban core districts consistently stabilized. In the period 2000 to 2005, the annual expansion rate of all the suburban areas grew fast, whereas the development of Qixia region became slow.

### 3.3. Dynamic Change in ISA in Different Directions

Figure 7 shows that Nanjing expansion showed a concentric circle characteristic. Compared with that in other cities, the overall density of ISA in Nanjing was lower in the same period. The reason was that Nanjing is a landscape city. The Yangtze River and Zijin Mountain are also in the middle of the city, thereby reducing the total density of ISA. From 1990 to 2017 within the distance of 0 to 5 km, the density of ISA was the highest in each period. The total ISA density in this area increased from 0.63 to 0.83. However, the density of ISA over 40 km was less than 0.3 in all the periods. At around 15 km, the ISA density drastically changed and increased from 0.10 in 1990 to 0.55 in 2017. This result also reflects that, before 2000, the main expansion mostly occurred in the city center in the range of 10 km. After 2000, especially after 2005, the city expansion concentrated on the space between 15 and 20 km.

Different direction figures show that a slight "U" appeared in the east, north, and northwest directions because of the topographic features, such as the Yangtze River and Zijin Mountain, which block the city expansion. However, another wave peak appeared after the U shape. This peak indicates that the urban subcenter was built around the 15 km area in each direction. The figure shows that the west side was not changing dramatically outside the 10 km area. The reason is that the west area belongs to the hilly area of Jianghuai and the urban construction is greatly influenced by the terrain. Figure 7 shows that the development of ISA in southwest Nanjing increased obviously around the 10 km area. This phenomenon is attributed to the formation of the new city center called Hexi. After 2000, the area between 10 and 25 km was influenced by the formation of the new city center. As the economy developed, the density of the 45 km area in southwest increased beyond the city concentric structure. The main reason is that the area between the two cities of the Yangtze River formed the economic zone. The Banqiao city belt is the economic belt between Nanjing and Maanshan. Three peaks in the south direction at 20, 55, and 80 km formed the southern New Town, Lukou Airport, and construction center of Gaochun District, respectively.

Fifteen and 50 km areas of the southeast formed the subcenter of Dongshan City and the construction center of Lishui District.

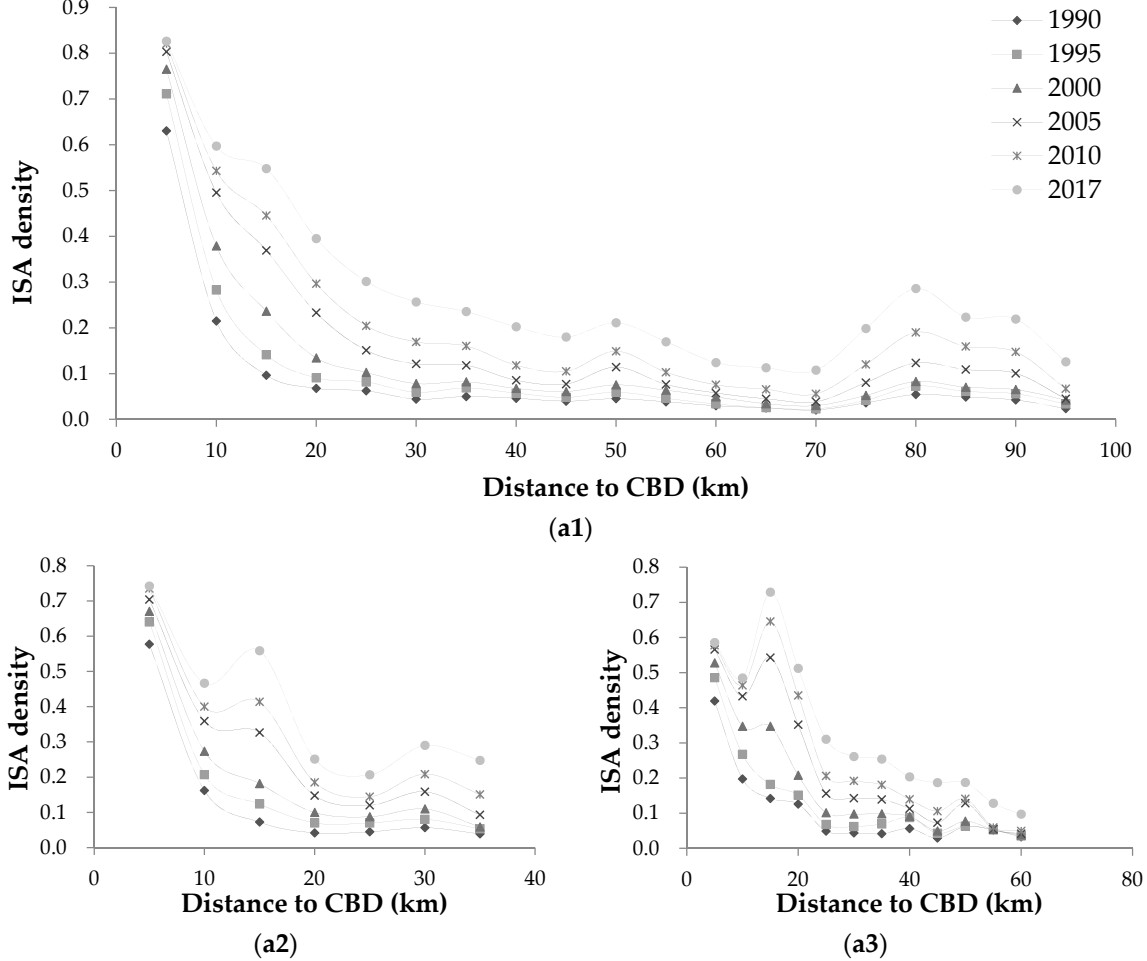

**Figure 7.** *Cont.*

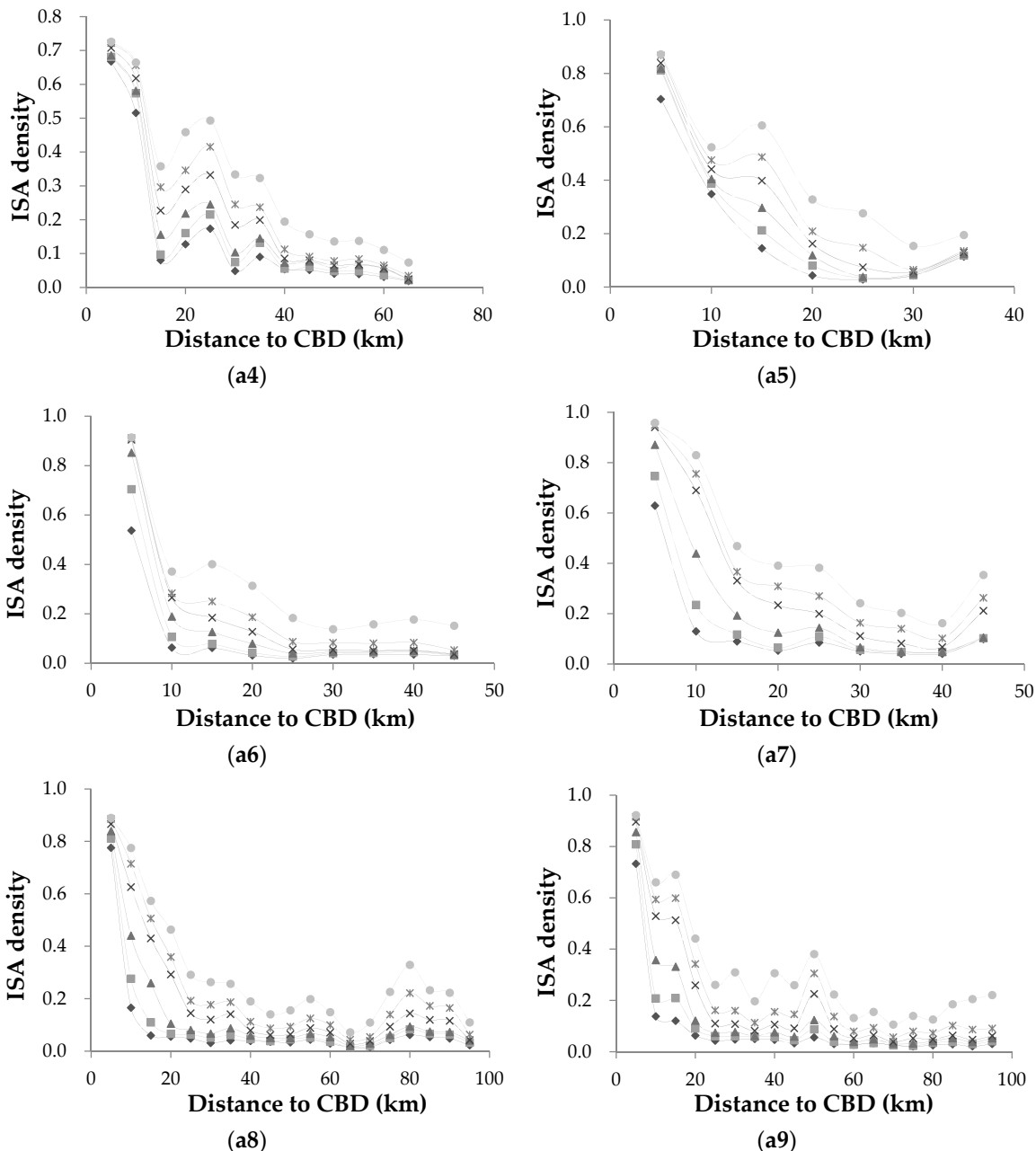

**Figure 7.** Density of ISA in different directions in Nanjing from 1990 to 2017. (Note: (**a1**) Entire area; (**a2**) East; (**a3**) Northeast; (**a4**) North; (**a5**) Northwest; (**a6**) West; (**a7**) Southwest; (**a8**) South; (**a9**) Southeast).

Analysis of the entire change of ISA density at different distances in Nanjing can reflect the focus area of city construction in different periods. Figure 8 indicates that, at the center of the city around 5 km, the change of ISA density was the largest in the period 1990 to 1995. Then, the rate gradually reduced to approximately 0.01 in 2010 to 2017. This finding shows that the city core formed in the period 1990 to 1995. After 1995, the focus of its construction shifted. Around the 10 km area, the ISA presented the greatest change rate in the period 2000 to 2005. After this period, the change rate of ISA decreased. Therefore, the region focused on its construction before 2005. Around the 15 km area, the change rate of ISA increased from 1990 to 2005 and reached the peak stage in the period 2000 to 2005. In the past years from 2010 to 2017, the change in ISA density was over 0.05 in the distance of 35 km area. Therefore, this trend is the focus of development and construction in recent years.

From Figure 8, the focus area of different directions can be analyzed. After 2000, the strength of construction increased in the 15 km area of the east. The high change value of the 30 to 35 km area in

2005 was due to the rapid construction of Tangshan according to the actual situation. In the northeast, the construction accelerated in the 15 km area in the period 1995 to 2005. At present, the construction focuses on the 25 km area. In the north, northwest, and west directions, the city grew very fast from 2010 to 2017. The main reason is the construction of Jiangbei national development zone and Hexi city core. After 2000, the change rate of ISA increased in the south, southeast, and southwest directions. The change rate of ISA has been the highest in recent years. Therefore, the south area is the main area of urban expansion after 2000.

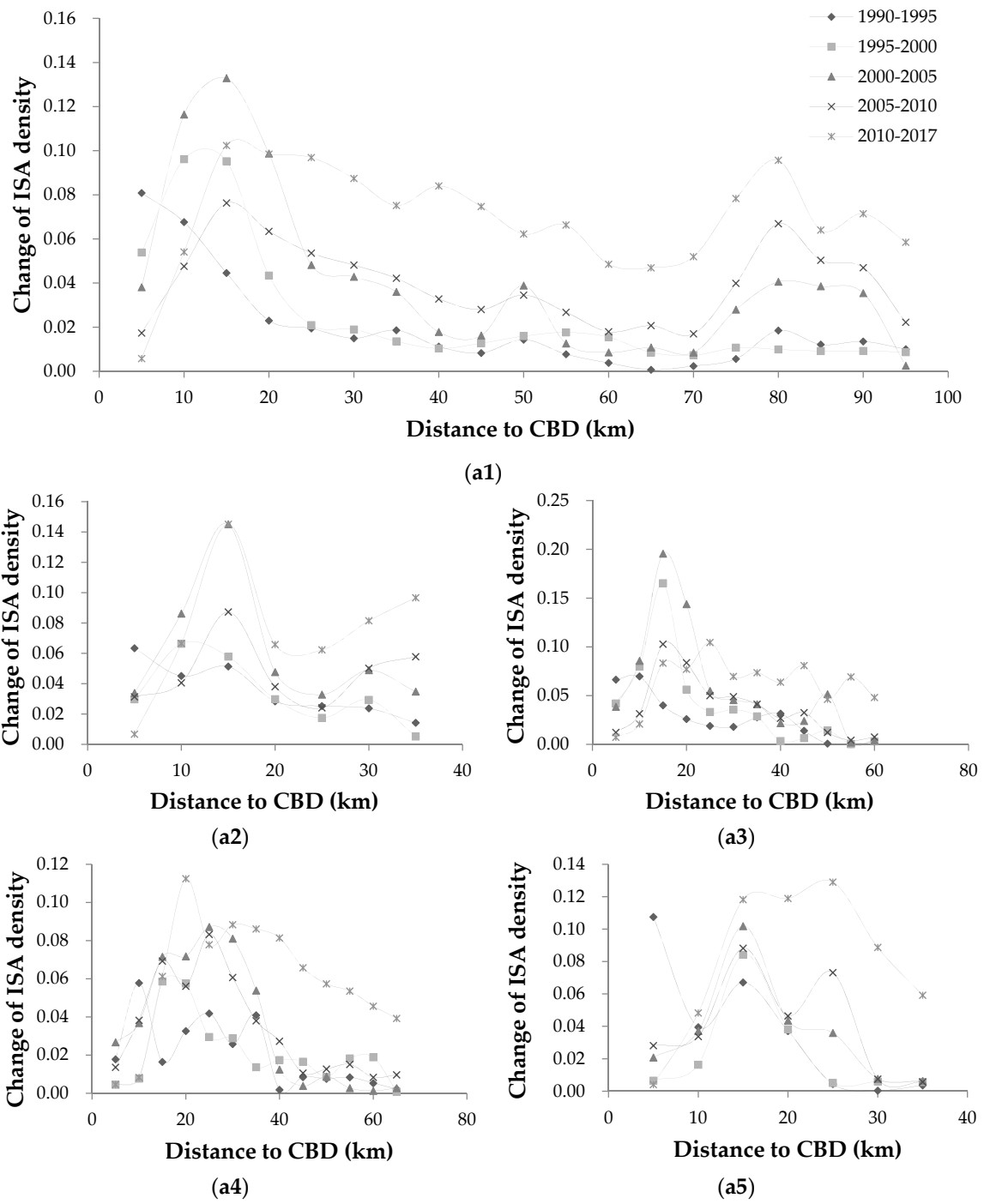

**Figure 8.** *Cont.*

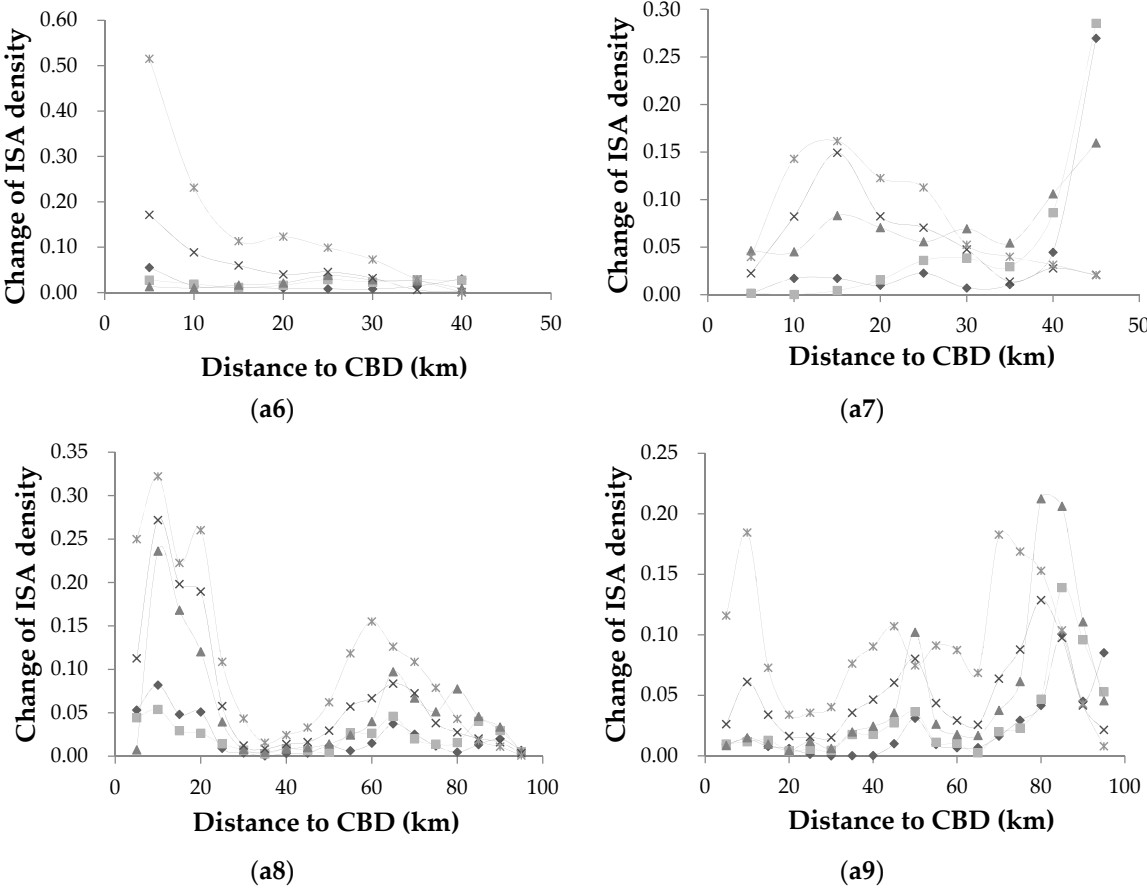

**Figure 8.** Density of change in ISA in different directions in Nanjing from 1990 to 2017. (Note: (**a1**) Entire area; (**a2**) East; (**a3**) Northeast; (**a4**) North; (**a5**) Northwest; (**a6**) West; (**a7**) Southwest; (**a8**) South; (**a9**) Southeast).

*3.4. Impacts of Topography on Dynamic Change in ISA*

Table 5 shows the ISA at different altitude levels from 1990 to 2017. The results show that most of the distribution of ISA was less than 40 m in height. In the 0 to 20 m and 20 to 40 m levels, the total amount increased from 198.71 (176.45 km$^2$) to 987.86 (629.37 km$^2$) from 1990 to 2017 as the altitude increased. The increase of ISA in the same period was small. That is, ISA increased in different altitudes. However, as altitude increased, the expansion rate became small. This result indicates that urban expansion occurs mainly in relatively low-altitude areas and that elevation is a constraint on the urbanization of Nanjing. A similar situation is observed for the slope groups. Table 6 shows that the majority of ISA was located in areas of slope values less than 5 degrees, and these areas were suitable for construction. This result indicates that the relationship between Nanjing expansion and terrain was obvious. The expansion of Nanjing was mainly distributed in the flat terrain, and the elevation and slope were important constraints for the expansion of the city (Figure 9).

**Table 5.** Impacts of elevation on dynamic change in ISA in Nanjing from 1990 to 2017.

|  | 0–20 | | 20–40 | | 40–60 | | 60–80 | | 80–100 | | 100–150 | | 150–200 | | >200 | |
|---|---|---|---|---|---|---|---|---|---|---|---|---|---|---|---|---|
|  | km$^2$ | % | km$^2$ | % | km$^2$ | % | km$^2$ | % | km$^2$ | % | km$^2$ | % | km$^2$ | % | km$^2$ | % |
| 1990 | 198.71 | 45.40 | 176.45 | 40.31 | 44.12 | 10.08 | 11.13 | 2.54 | 3.80 | 0.87 | 2.98 | 0.68 | 0.46 | 0.11 | 0.05 | 0.01 |
| 1995 | 251.78 | 45.05 | 227.93 | 40.78 | 54.59 | 9.77 | 14.67 | 2.62 | 5.27 | 0.94 | 3.88 | 0.69 | 0.66 | 0.12 | 0.08 | 0.01 |
| 2000 | 329.59 | 48.14 | 268.42 | 39.21 | 61.05 | 8.92 | 15.08 | 2.20 | 5.42 | 0.79 | 4.03 | 0.59 | 0.92 | 0.13 | 0.14 | 0.02 |
| 2005 | 518.78 | 51.32 | 371.43 | 36.74 | 83.84 | 8.29 | 20.61 | 2.04 | 7.77 | 0.77 | 6.35 | 0.63 | 1.79 | 0.18 | 0.40 | 0.04 |
| 2010 | 689.58 | 52.96 | 463.18 | 35.57 | 104.94 | 8.06 | 24.42 | 1.88 | 9.12 | 0.70 | 7.85 | 0.60 | 2.34 | 0.18 | 0.59 | 0.05 |
| 2017 | 987.86 | 54.19 | 629.37 | 34.53 | 147.60 | 8.10 | 32.44 | 1.78 | 11.62 | 0.64 | 9.86 | 0.54 | 3.13 | 0.17 | 1.02 | 0.06 |

**Table 6.** Impacts of slope on dynamic change in ISA in Nanjing from 1990 to 2017.

|  | 0–5 | | 5–10 | | 10–15 | | 15–20 | | >20 | |
|---|---|---|---|---|---|---|---|---|---|---|
|  | km$^2$ | % | km$^2$ | % | km$^2$ | % | km$^2$ | % | km$^2$ | % |
| 1985 | 121.72 | 41.67 | 119.44 | 40.89 | 38.11 | 13.05 | 9.16 | 3.14 | 3.68 | 1.26 |
| 1990 | 184.09 | 43.01 | 173.68 | 40.57 | 52.39 | 12.24 | 12.45 | 2.91 | 5.44 | 1.27 |
| 1995 | 235.1 | 43.03 | 222.24 | 40.67 | 66.75 | 12.22 | 15.59 | 2.85 | 6.74 | 1.23 |
| 2000 | 294.18 | 43.93 | 270.89 | 40.45 | 79.18 | 11.82 | 18.03 | 2.69 | 7.43 | 1.11 |
| 2005 | 446.13 | 45.10 | 395.2 | 39.95 | 111.48 | 11.27 | 25.11 | 2.54 | 11.21 | 1.13 |
| 2010 | 585.97 | 45.98 | 504.69 | 39.61 | 138.13 | 10.84 | 30.95 | 2.43 | 14.55 | 1.14 |
| 2017 | 836.25 | 46.88 | 703.1 | 39.41 | 185.66 | 10.41 | 40.27 | 2.26 | 18.65 | 1.05 |

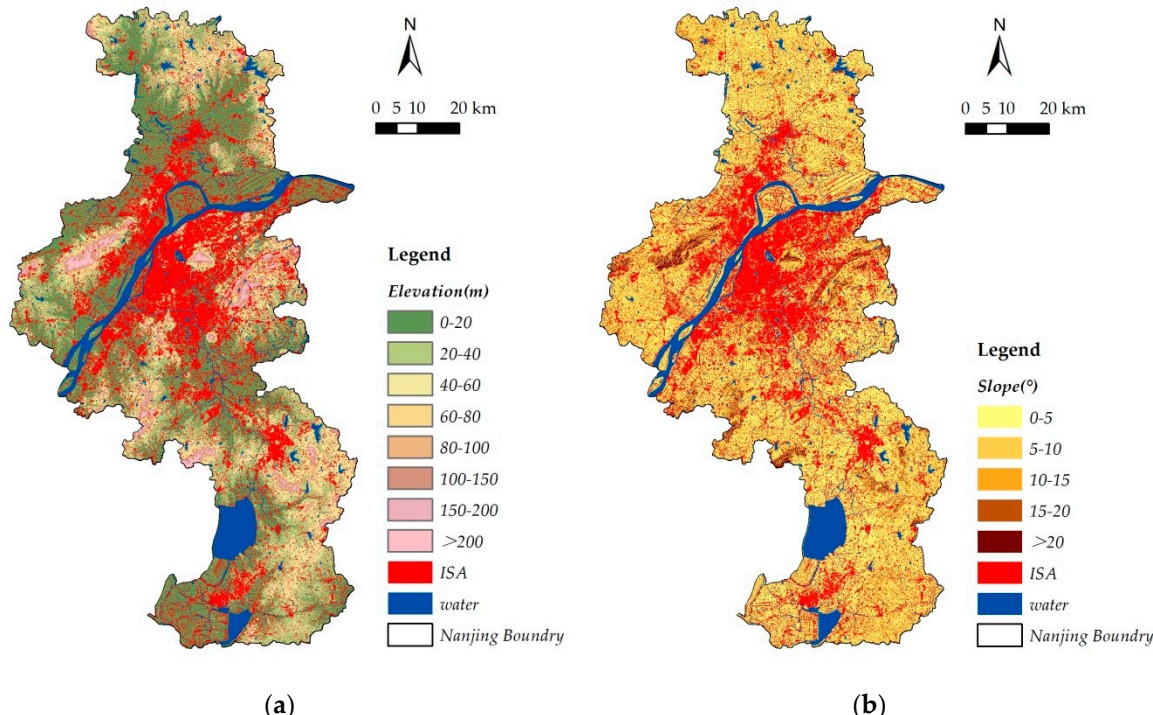

(**a**) (**b**)

**Figure 9.** Impacts of Topography in ISA. (Note: (**a**) Elevation; (**b**) Slope).

*3.5. Relationship between ISA Dynamics and GDP in Different Districts*

Economic condition is an important driver of urbanization. In this study, we examined the relationship between GDP and ISA by exploring regression models of 11 districts in Nanjing. The result shows a strong positive relation between ISA and GDP at different periods; the coefficients of determination ($R^2$) reached more than 0.88 for all districts (Table 7).

Figure 10 shows that GDP was more sensitive to ISA in the suburban districts than the urban core districts in Nanjing. In the urban core districts, the changes in ISA in the Jianye and Yuhuatai areas were sensitive to the changes in GDP. Compared with other mature urban core districts, the changes in ISA in Jianye and Yuhuatai presented a great development space in the next few years. In the suburb districts, the driving force of Qixia's economic development to ISA was the smallest, followed by that of Jiangning. The ISA of Pukou, Luhe, and Gaochun was significantly affected by GDP, and the state of the total ISA increased significantly.

**Table 7.** Relationship between ISA dynamics and GDP in different districts in Nanjing.

| District | Formula | $R^2$ |
|---|---|---|
| Xuanwu | y = 2.9239 ln(x) + 16.24 | 0.9672 |
| Qinhuai | y = 2.8946 ln(x) + 20.198 | 0.9747 |
| Gulou | y = 0.9895 ln(x) + 37.588 | 0.9711 |
| Yuhuatai | y = 10.965 ln(x) - 10.446 | 0.9864 |
| Jianye | y = 8.5131 ln(x) + 11.01 | 0.9683 |
| Qixia | y = 26.873 ln(x) − 16.089 | 0.9636 |
| Gaochun | y = 38.893 ln(x) − 82.051 | 0.9259 |
| Pukou | y = 44.196 ln(x) − 92.63 | 0.8891 |
| Luhe | y = 54.399 ln(x) − 91.679 | 0.9044 |
| Lishui | y = 48.067 ln(x) − 119.11 | 0.9529 |
| Jiangning | y = 104.21 ln(x) − 312.02 | 0.9828 |

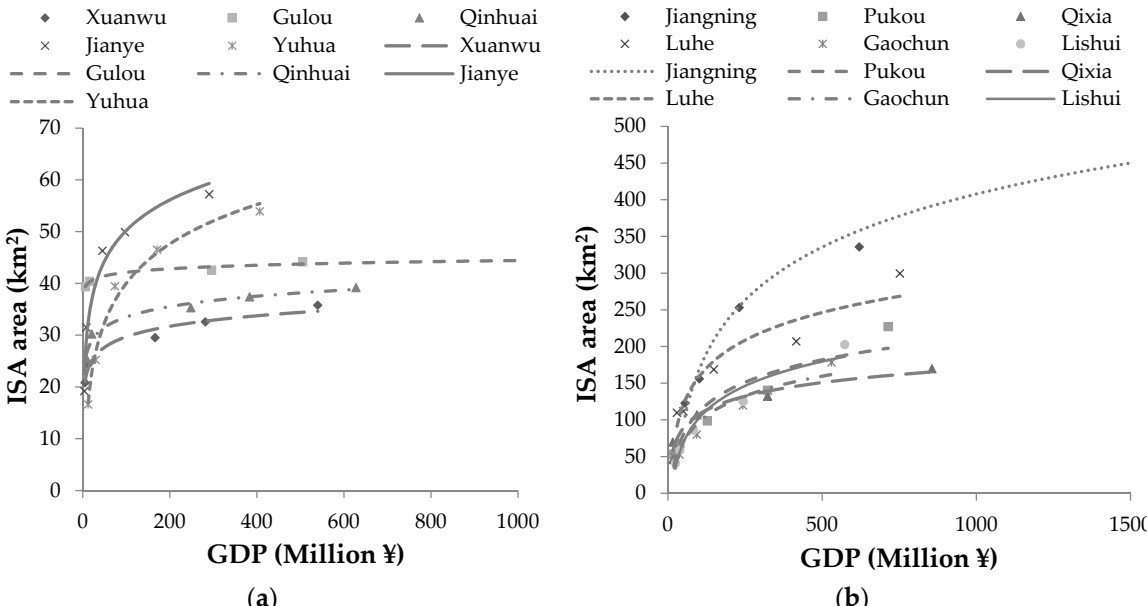

**Figure 10.** Relationships between ISA and GDP in different directions in Nanjing from 1990 to 2017. (Note: (**a**) Relationships between ISA and GDP in urban core districts; (**b**) Relationships between ISA and GDP in suburban districts).

## 4. Discussion

### 4.1. Improving the Accuracy of Urban Expansion Research Data

Urban expansion analysis has been usually conducted by extracting the data of urban construction land use [14–17]. However, the relatively coarse spatial resolution of Landsat images, mixed pixel, and complex urban land cover composition reduces the accuracy of urban construction land classification [19]. In this study, the comprehensive method of a pixel-based classification was used to extract ISA. This method is more accurate than the construction land that is directly extracted from Landsat files. We improved LSMA, which has been validated by scholars as having a valid ISA extraction accuracy, through a variety of methods, such as Modified Normalized Difference Water Index (MNDWI) and decision tree. The improved method can effectively extract ISA with an overall accuracy of 90%.

This method is effective in producing accurate distributions of ISA, which is especially necessary for testing urban expansion. Given that this study explores the expansion of land use in each district and different directions in Nanjing, the improvement in the accuracy of ISA plays a very crucial role in

the entire study. Accurate extraction of ISA allows a clear analysis of ISA changes in each area and direction of the urban expansion.

*4.2. Analyzing City Expansions at Different Spatial Scales and Direction Levels*

Rational control of the pace and direction of the expansion of construction land in metropolitan areas has been the focus of Chinese urbanization and land management policies. Research on the expansion mechanism of construction land through different directions and spatial scales can provide a scientific basis for urban development management.

This study established a relatively comprehensive analysis system of urban expansion. From the macroscopic view of the overall urban expansion, the density changes, and center of gravity of ISA were judged, and, thus, the overall trend of urban expansion was analyzed. From the perspective of micro-regional and directional levels, we studied the expansion process and change trend of ISA in every district and direction of Nanjing and analyzed the key areas and directions of urban expansion in the future. The date base of multi-center development in Nanjing can be provided for urban planning and construction needs.

On the basis of this systematic research model and institutional analysis of the expansion of construction land use in Nanjing, additional research experience and resources can be provided. The results can be used to summarize a set of characteristics suitable for Chinese urbanization and land expansion construction.

*4.3. Effects of Topographic and GDP Factors on Urban Expansion*

Topography and GDP are two important factors affecting urban ISA expansion in Nanjing. This research confirmed this fact. Examining the relationships between topography and ISA at different periods shows that topography is still an important constraint factor affecting the expansion of the city even with developed engineering technology.

Using regression analysis parsing the relationship between ISA and GDP in different districts in Nanjing indicates that ISA is positively related to GDP, especially in districts at the development stage. GDP in most of these districts significantly influences urban expansion. Those districts are usually located on the urban fringe area. By contrast, GDP in other types of districts insignificantly influences the ISA expansion.

On the basis of the analysis of topography and economy, the future development of different districts in Nanjing can be analyzed and the current and future development focuses of different areas can be summarized. Effective strategies for urban planning, policy and improved service for urban citizens can be provided.

*4.4. Social and Political Factors on Urban Expansion*

Social and political factors are also important factors affecting ISA expansion in Nanjing. The household registration system and the reform and opening up policies are the main factors affecting the expansion of Chinese cities in the 21st century.

In 1958, China launched the household registration system, which divided the identity of urban residents and rural residents, imposed strict restrictions on the free movement of people [55]. Since the 1990s, with the reform of the household registration system, the urban and rural residents have flowed with each other, but the inequality of resources between urban and rural areas has caused the population concentrated in the central urban area which can provide the relatively-perfected educational and medical conditions. Many rural residents have flocked to the cities, and it indirectly led to the emergence of a basic pattern of city expansion in China, and Nanjing is no exception. In recent years, to attract talents and improve the competitiveness of Nanjing, the local government has formulated a new policy that people with the bachelor's degree or above can be directly settled down, the possibility of sustainable expansion of the city's scale has been provided.

Since the reform and opening up, Chinese society has entered a new stage of transformation and development [56]. Under the guidance of market economic policy, Chinese enterprises have begun to change from government-led enterprises to market-oriented enterprises. Meanwhile, the hot spots of foreign investment in China are the Yangtze River Delta and the Pearl River Delta, and Nanjing, as a hot investment city in the Yangtze River Delta, has accelerated its urban expansion due to the influx of large quantities of local and foreign capital, and it is embodied in the agglomeration development mode with industrial parks. Those industrial parks have brought the agglomeration of industry and population, accelerated the development of cities, and promoted the change of urban spatial structure.

## 5. Conclusions

Compared with other European and American megacities, Chinese megacities exhibit different urbanization processes, patterns, and mechanisms. This study used Nanjing as a case study to assess and analyze urban expansion by using a comprehensive method to extract and analyze ISA.

From the results, the following conclusions were drawn:

(1) The proposed comprehensive method can effectively improve the accuracy of ISA extraction. In this case study, the overall accuracy of ISA in Nanjing was over 90%. The ISA of the city was comprehensively analyzed by macro analysis of urban expansion and microcosmic analysis of ISA in different districts and directions.

(2) Urban development of Nanjing has been expanding rapidly, slowing down, and expanding rapidly again. The annual expansion area from 23.64 $km^2$/year in 1990 to 1995 increased to 63.77 $km^2$/year in 2000 to 2005, slowed down in 2005 to 2010, and then increased again in 2010 to 2017 with the annual expansion area of 72.61 $km^2$/year. The region of high ISA density continues to grow. The center of gravity of ISA obviously moves southeast, and the overall direction of expansion is relatively balanced in recent years.

(3) Types and trends of ISA changes in urban core and suburban districts obviously differ. In recent years, the increase in the area and density of ISA in the urban core districts is relatively slow, and the change in ISA in the suburban districts is the opposite. At the same time, the different areas and rates of ISA change from the different directions of urban CBD and are greatly influenced by the urban development policy and geographical factors.

(4) The case of Nanjing indicates that topography and economy are important influencing factors of urbanization. In different districts at dissimilar stages of development, the changes and trends of ISA differ. The increase of the ISA in the districts at the stage of development is positively proportional to GDP. Based on the regression analysis of the economy and ISA, the future expansion trends of each district can be approximately estimated.

(5) Urban expansion in Nanjing represents a common phenomenon in the development of the second-tier cities in China. The ISA analysis of Nanjing's expansion indicates that Nanjing is still in a period of rapid urban expansion, the ISA density of the original urban central district continues to increase, and the overall ISA area of the city continues to expand. These are also the basic characteristics of the expansion of Nanjing and the other second-tier cities in China. Meanwhile, Nanjing's urban expansion can provide some development experience for the same type of second-tier cities and the third-tier cities in China, and the government can formulate corresponding planning strategies according to the development of the ISA distribution and the future ISA expansion trends, analyze the urban spatial form, improve the level of urban management, and make rational use of urban space.

**Author Contributions:** Z.W. conceived and designed the experiments; Y.Q. made contributions to the design, data processing; Z.W. wrote the paper; All authors had read and approved the manuscript.

**Acknowledgments:** The present study is supported by the National Natural Science Foundation of China (Grant No. 31570703). The authors are grateful for the technical support of Professor Shiguang Shen.

**Conflicts of Interest:** The authors declare no conflict of interest.

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
