# Peer review of "Study on Urban Expansion Using the Spatial and Temporal Dynamic Changes in the Impervious Surface in Nanjing"

_sustainability, doi:10.3390/su11030933_

Round 1

Reviewer 1 Report

The study is interesting. It focuses on an important Chinese city, affected by an accelerated urban growth in recent times. From the technical point of view: the analysis of urban growth, total and according to different variables, the result is satisfactory. The techniques of measurement and commentary are perfectly mastered.
However, there are very important gaps to be solved. What social and political factors justify urban growth? They are not commented. What is the existing residential inequality?
In addition, what lessons can be learned from Nanjing to learn about Chinese urban dynamics today? Although the last pages and in the coclusions are suggested, what transferability of the results obtained in Nanjung for other cases?
These elements must be contemplated to accept the publication.

Author Response

Response to Reviewer 1 Comments

Point 1: What social and political factors justify urban growth? They are not commented. What is the existing residential inequality?

Response 1: Thank you for your advice. This article mainly focuses on Nanjing city research through the ISA extraction, the intention focuses on the analysis through the data quantification. Since social and political factors are not easy to quantify, they have not been mentioned in this paper before.

However, social and political factors are the important factors affecting urban development and expansion. The suggestion is very valuable for me, according to the original article structure, I added this to the discussion part and added “4.4. Social and Political Factors on Urban Expansion”.

In this social and policy political factors, the household registration system and the reform and opening up policies were mentioned, they are two main factors affecting the expansion of Chinese cities in the 21st century.

At the same time, this part also explains the unequal situation of education, medical and other resources, and analyses the impact of this inequality on urban expansion.

Line 388-409

“4.4. Social and Political Factors on Urban Expansion

Social and Political Factors are also important factors affecting ISA expansion in Nanjing. The household registration system and the reform and opening up policies are the main factors affecting the expansion of Chinese cities in the 21st century.

In 1958, China launched the household registration system, which divided the identity of urban residents and rural residents and imposed strict restrictions on the free movement of people[55]. Since the 1990s, with the reform of the household registration system, the urban and rural residents have flowed with each other, but the inequality of resources between urban and rural areas have caused the population concentrated in the urban central where can provide the relatively-perfected educational and medical conditions. Many rural residents have flocked to the cities, it indirectly led to the emergence of a basic pattern of city expanding in China, and Nanjing is no exception. In recent years, in order to attract talents and improve the competitiveness of Nanjing, the local government has formulated a new policy that people with the bachelor's degree or above can be directly settled down, the possibility of sustainable expansion of the city's scale has been provided.

Since the reform and opening up, Chinese society has entered a new stage of transformation and development[56]. Under the guidance of market economic policy, Chinese enterprises have begun to change from government-led enterprises to market-oriented enterprises. Meanwhile, the hot spots of foreign investment in China are the Yangtze River Delta and the Pearl River Delta, Nanjing, as a hot investment city in the Yangtze River Delta, has accelerated its urban expansion due to the influx of large quantities of local and foreign capital, and it is embodied in the agglomeration development mode with industrial parks. Those industrial parks have brought the agglomeration of industry and population, accelerated the development of cities, and promoted the change of urban spatial structure.”

Point 2: what lessons can be learned from Nanjing to learn about Chinese urban dynamics today? Although the last pages and in the coclusions are suggested, what transferability of the results obtained in Nanjing for other cases?

Response 2:

As a second-tier city in China, urban expansion in Nanjing represents a common phenomenon in the development of these cities. Through the case of Nanjing, there are two aspects can be learned, one is topography and economy are important influencing factors of urbanization, the other is ISA density of the original urban central district continues increasing, and the overall ISA area of the city continues expanding are two basic characteristics of the expansion of Nanjing and the other second-tier cities in China. Meanwhile, these characteristics can be used as a reference for urban planning strategies.

Line 430-444

 “(4) The case of Nanjing indicates that topography and economy are important influencing factors of urbanization. In different districts at dissimilar stages of development, the changes and trends of ISA differ. The increase in the ISA in the districts at the stage of development is positively proportional to GDP. Based on the regression analysis of the economy and ISA, the future expansion trends of each district can be approximately estimated.

(5) Urban expansion in Nanjing represents a common phenomenon in the development of the second-tier cities in China. The ISA analysis of Nanjing's expansion indicates that Nanjing is still in a period of rapid urban expansion, the ISA density of the original urban central district continues increasing, and the overall ISA area of the city continues expanding. These are also the basic characteristics of the expansion of second-tier cities in China. Meanwhile, Nanjing's urban expansion can provide some development experience for the same type of second-tier cities and the third-tier cities in China, the government can formulate corresponding planning strategies according to the development of the ISA distribution and the future ISA expansion trends, analyze the urban spatial form, improve the level of urban management, and make rational use of urban space.”

Reviewer 2 Report

Will you revise the sentences marked by the comments box adn the yellow colour.

Author Response

Response to Reviewer 2 Comments

Point 1: Title: "Study on urban expansion using the spatial and temporal dynamic changes" is better.

Response 1: Thank you for your advice, I changed the title into “Study on urban expansion using the spatial and temporal dynamic changes in the impervious surface in Nanjing”, please see the detail of the new title.

Point 2: L12: "timely and accurately for " These adverb words should be moved after the verb of analysis.

Response 2: Yes, I moved "timely and accurately for" after the verb of analysis

Line 12-13

“It is of great significance to analysis timely and accurately the dynamic changes of impervious surface for urban development planning.”

Point 3: L14: "the time series data on" is better.

Response 3: Thanks for your suggestion, I added "data on" into the sentence.

Line 14

“In this study, we use a comprehensive method to extract the time series data on impervious surface area (ISA) from the multi-temporal Landsat remote sensing images with a high overall accuracy of 90%”.

Point 4: L16: You need to add some words to explain “administration and direction” more clearly, such as "political administration and direction level" or "level of the government".

Response 4: Thanks for your suggestion, I changed the word into “political administration and direction level”.

Line16

“The processes and mechanisms of urban expansion at different political administration and direction level in Nanjing metropolitan area are investigated using the comprehensive classification method consisting of minimum noise fraction, linear spectral mixture analysis, spectral index, and decision tree classifiers.”

Point 5: L23: “Gravity” is not adequate. "population" or other word is better.

Response 5: Thanks for reviewer’s comments. The center of gravity is a geometric property of the city. The center of gravity is the average location of the weight of the city. In order to describe it more clearly, I change the sentence into.

Line 23

“the geometric gravity center of construction land”

Point 6: L69: Will you explain the definition of center of gravity.

Response 6: The center of gravity is a geometric property of the city. The center of gravity is the average location of the weight of the city.

Point 7:  L89: Figure 1: Will you revise the unclear curve along the coast line.

Response 7: Yes, I revised the figure, please see the detail on Figure 1.

Point 8: L89: Figure 1: Will you revise the unclear part along the east border.

Response 8: Yes, I revised the figure, please see the detail on Figure 1. (the same pic above)

Point 9:  L94: Will you add some words to explain what kind of shape file.

Response 9: Yes, I added "The Environmental Systems Research Institute (ESRI) Shapefile" before "shapefile".

Line 95

“The Environmental Systems Research Institute (ESRI) Shapefile of Nanjing administrative units was used.”

 Point 10: L96: Will you add the words to explain Geogle Earth images of which year taken.

Response 10: Thanks for comments, and I added "from 1990, 1995, 2000, 2005, 2010, and 2017" after "Geogle Earth images".

Line 97

“Google Earth images from 1990, 1995, 2000, 2005, 2010, and 2017 were used as the reference data for evaluating ISA accuracy”.

Point 11: L120: Will you move this word to the center position.

Response 11: Thanks for comments, but the example form is that.

Point 12: L146: Will you use the capital letter of these words, as follows, Central Business District.

Response 12: Yes, I use the capital letters of "Central Business District"

Line149

“This model indicates that a city takes the Central Business District (CBD) as its expansion circle center and expands from the center to suburbs in the form of concentric circles.”

Point 13: L174: Will you explain what kind of accuracy assessment was done, and the definition of the value of accuracy.

Response 13: Thank you for your question. The evaluation method has been accurately written in the text.

Line176

“The accuracy assessment of confusion matrix show that the overall accuracy of the ISA data in Nanjing was more than 90% in 2010 and 2017 (Table 2). ”

This evaluation method is often used in remote sensing, so there is no detailed explanation in the text. The specific definition is as follows.

①overall accuracy

 It is equal to the sum of the correctly classified pixels divided by the total number of pixels.

②Producer’s accuracy

It refers to the ratio of the number of pixels (diagonal values) correctly classified by the classifier into category A and the total number of category A (the sum of category A columns in the confusion matrix).

③User’s accuracy

It refers to the ratio of the total number of pixels (diagonal values) correctly classified into category A and the total number of pixels (the sum of rows in category A in the confusion matrix) that the classifier classifies the whole image into categories A.

Point 14: L316: 5 deg is better.

Response 14: Thanks for your good suggestion; I changed "°" to "deg".

Line156 and L319.

“22.5deg” and “5deg”

Point 15: L386: If you mean this word of Western to be European and American, you had better to revise this word.

Response 15: Thanks for your question; I revised it. to

Line 411

“Compared with other European and American megacities”.

Reviewer 3 Report

The purpose of the manuscript is to examine urban expansion based on various impervious surface area indexes and concentric and regression analysis. The paper is well organized with good technical quality and data whereas it adds some new information’s to the existing body of knowledge in the field of improving the accuracy of urban expansion data. However, some minor points have to be addressed:

           - Line 33: The authors could also add that the urbanization is effecting the storm hydrograph, sediment yield and flood hazard too as it is documented at the following recent articles: Myronidis D., Ioannou K. 2019: Forecasting the Urban Expansion Effects on the Design Storm Hydrograph and Sediment Yield Using Artificial Neural Networks, Water 2019, 11, 31; doi:10.3390/w11010031, Myronidis D., Stathis D., Sapountzis M. 2016: Post-Evaluation of Flood Hazards Induced by Former Artificial Interventions along a Coastal Mediterranean Settlement, Journal of Hydrologic Engineering, 21(10), 05016022,

           - Line 74: The added valued of this manuscript in the existing body of knowledge should be clearly stated.

           - Line 84: A reference of figure 1 must be placed here and not on line 156

           - Figure1: The Yangtze River could be added on Figure 1.

           - Line 93: It is Table 1.

           - Equation (3), (4) and (5): The authors should explain what the t1 and t2 variables are

           - Line 177 a reference of Table 2 must be added here.

-       In each citation it should be added the DOI. For example: Dupras, J.; Marull, J.; Parcerisas, L.; Coll, F.; Gonzalez, A.; Girard, M.; Tello, E. The impacts of urban sprawl on ecological connectivity in the montreal metropolitan region. Environmental Science & Policy  2016, 58, 61-73. DOI: 10.1016/j.envsci.2016.01.005

Author Response

Response to Reviewer 3 Comments

Point 1: Line 33: The authors could also add that the urbanization is effecting the storm hydrograph, sediment yield and flood hazard too as it is documented at the following recent articles: Myronidis D., Ioannou K. 2019: Forecasting the Urban Expansion Effects on the Design Storm Hydrograph and Sediment Yield Using Artificial Neural Networks, Water 2019, 11, 31; doi:10.3390/w11010031, Myronidis D., Stathis D., Sapountzis M. 2016: Post-Evaluation of Flood Hazards Induced by Former Artificial Interventions along a Coastal Mediterranean Settlement, Journal of Hydrologic Engineering, 21(10), 05016022

Response 1: Thanks for your good suggestion, and I have added these two papers to my manuscript, please see the reference 12 and 13, and mentioned the transformation of the earth’s surface from non-urban land to urban land may cause flood hazard and so on.

Point 2:  Line 74: The added valued of this manuscript in the existing body of knowledge should be clearly stated.

Response 2: Thanks for your question, I added two points of the added valued of this manuscript in the existing body of knowledge. One is providing a method to improve the accuracy of urban expansion research data extracted form RS images, the other is to better understand the details of urban expansion in Nanjing.

Line 73 to Line 78.

This type of research provides a method to improve the accuracy of urban expansion research data extracted form RS images and provide new understanding on the interactions of socioeconomic systems, different development phases, expansion mechanisms, and urbanization patterns from the perspective of direction and district levels and the entire city. The other value of this study is to better understand the details of urban expansion in Nanjing, and to provide valuable insights for improving urban planning, management, and sustainability.”

Point 3:  Line 84: A reference of figure 1 must be placed here and not on line 156

Response 3: Thanks for your suggestion. I added "(Figure 1)" on former Line84(now Line 86).

Line 86

 Nanjing metropolis includes 11 districts, namely, Xuanwu, Qinhuai, Gulou, Jianye, Yuhuatai, Qixia, Pukou, Luhe, Jiangning, Lishui, and Gaochun, with a total area of 6587.02 km2[45] (Figure 1) .

Point 4:  The Yangtze River could be added on Figure 1.

Response 4: Thanks for your advice. I deleted the Yangtze River part from Figure 1. Please see the detai on Figure 1.

Figure 1

Point 5:  Line 93: It is Table 1.

Response 5: Thanks for your question, I revised "Table 2" to "Table 1".

Line94

“The RS data used in this research were mainly from the Landsat Thematic Mapper (TM), Enhanced ETM+, and Operational Land Imager from 1990, 1995, 2000, 2005, 2010, and 2017 (Path: 120, Row: 38) (Table 1).”

Point 6:  Equation (3), (4) and (5): The authors should explain what the t1 and t2 variables are.

Response 6: Thanks for your question. I added "Where ISA(t) is the area of ISA in different time, t1 is the prior time of t2" to write explain what the t1 and t2 variables are. Please see the detail on L146.

Point 7:  Line 177 a reference of Table 2 must be added here.

Response 7: Thanks for your suggestion. I added "(Table 2)" on Line177.

Line177.

“The accuracy assessment of confusion matrix show that the overall accuracy of the ISA data in Nanjing was more than 90% in 2010 and 2017 (Table 2)”.

Point 8:  In each citation it should be added the DOI. For example: Dupras, J.; Marull, J.; Parcerisas, L.; Coll, F.; Gonzalez, A.; Girard, M.; Tello, E. The impacts of urban sprawl on ecological connectivity in the montreal metropolitan region. Environmental Science & Policy  2016, 58, 61-73. DOI: 10.1016/j.envsci.2016.01.005

Response 8: Thanks for your question; I read some recently papers publish on journal of SUSTAINABILITY, and the reference didn’t show the DOI. And the example I have download form the website also didn’t show the DOI, So I didn’t add it. If it is needed, I will add it.

Round 2

Reviewer 1 Report

The authors have responded to many of the improvement comments made in the previous review. They have corrected the problems detected. They have even incorporated a new section in the text, on the sociopolitical context of urban growth in Nanjing and in China in general. The bibliography has improved.
The result is satisfactory for the publication.

Author Response

Response to Reviewer 1 Comments

Point 1: The authors have responded to many of the improvement comments made in the previous review. They have corrected the problems detected. They have even incorporated a new section in the text, on the sociopolitical context of urban growth in Nanjing and in China in general. The bibliography has improved.
The result is satisfactory for the publication.

Response 1: Thank you for your patient work. I took some time to revise my manuscript and uploaded revision paper. Some grammatical errors in the original text have been corrected, using the "Track Changes" function in Microsoft Word. Thank you very much, the suggestion is very valuable. I appreciate your kindly remind and help. Best wishes.
